# A New Invasive Pest in Plums: Monitoring and Control of *Eurytoma schreineri*

**DOI:** 10.3390/insects16111112

**Published:** 2025-10-31

**Authors:** Jana Ouředníčková, Michal Skalský, Martin Mészáros

**Affiliations:** Research and Breeding Institute of Pomology Holovousy Ltd., 508 01 Holovousy, Czech Republic; michal.skalsky@vsuo.cz (M.S.); martin.meszaros@vsuo.cz (M.M.)

**Keywords:** plum seed wasp, sticky traps, pesticides, IPM, fruit protection

## Abstract

**Simple Summary:**

The plum seed wasp *Eurytoma schreineri* is an invasive pest recently detected in the Czech Republic, posing a growing threat to European plum production. The larvae develop inside plum stones, causing premature fruit drop and yield losses of up to 90%. This study aimed to develop practical monitoring and control methods suitable for integrated pest management (IPM). Field experiments tested the attractiveness of differently coloured sticky traps, while laboratory and orchard trials evaluated the efficacy of several insecticides permitted in IPM systems. Results showed that yellow sticky traps were the most attractive to adult wasps and can serve as an effective tool for detecting pest presence and timing control measures. Among the tested insecticides, spinosad showed the highest efficacy under both laboratory and field conditions, reducing fruit infestation from nearly 80% in untreated orchards to 22% in treated plots. The combination of yellow trap monitoring and targeted use of spinosad represents an effective and environmentally acceptable approach to reducing *E. schreineri* populations. These findings provide growers with new tools for early detection and sustainable control of this invasive pest in plum orchards.

**Abstract:**

*Eurytoma schreineri* is a new invasive pest in the Czech Republic and may pose a risk in other European countries. Fruit damage caused by *E. schreineri* results in premature fruit drop, sometimes up to 90%. This study is the first to test the attractiveness of differently coloured sticky traps as a monitoring tool to determine presence, abundance, and timing of treatments against adults. We also evaluated efficacy of insecticides allowed in integrated pest management (IPM). Laboratory and field trials demonstrated that yellow sticky traps were the most attractive. Spinosad proved to be the most effective insecticide under both laboratory and orchard conditions, reducing infestation to 22.2% compared to 79.2% in untreated orchards. These findings provide new knowledge on pest monitoring and control strategies and will support fruit growers in developing effective IPM systems against *E. schreineri*.

## 1. Introduction

The plum seed wasp *Eurytoma schreineri* (Hymenoptera; Eurytomidae) is a new invasive (non-native) pest species in Central Europe. *Eurytoma schreineri* is native to the European part of Russia. *Eurytoma schreineri* was first described by Schreiner (1908) as a pest of the plum *Prunus domestica* (Linnaeus) in the region of Astrachan [1]. In the second half of the last century, the species spread to Moldova [2], Romania, and Turkey [3,4]. The occurrence of *E. schreineri* has been reported in Greece [5] and also in Western Siberia [6]. This species was recorded for the first time in Bulgaria and in the Czech Republic in 2013 [6,7]. Thus, according to this available information, it is spreading from the east and there is a risk of further expansion to the western part of Europe.

*Eurytoma schreineri* is a relatively new pest in plum orchards. We did not have much information on this insect. Therefore, the main objective was to build up a methodology for monitoring and controlling of *E. schreineri*. This was achieved by determining the most attractive visual trap as a simple tool to determine the presence, abundance, and optimal duration of treatment against *E. schreineri* adults. Another goal was to find an effective insecticidal treatment allowed in IPM that would prevent damage caused by *E. schreineri*. Insecticidal treatment against adults of *E. schreineri* is currently the main option for protecting plums from this dangerous invasive pest.

### 1.1. Host Plants

This invasive pest mainly attacks plums *P. domestica* and *P. domestica insititia* (ssp.) (Schneid). Preferred species are mirabelle plum, *P. domestica syriaca* (Borkhausen), myrobalan, *P. cerasifera* (Ehrhart), and apricot, *P. armeniaca* (Linnaeus). Blackthorn, *P. spinosa* (Linnaeus) is less preferred. Occurrence on cherry, *P. avium* (Linnaeus) and sour cherry, *P. cerasus* (Linnaeus) has not been confirmed [6].

### 1.2. Insect Description

*Eurytoma schreineri* belongs to the order Hymenoptera, suborder Clistogastra, superfamily Chalcidoidea, family Eurytomidae [1]. The adult body is black. Wings are transparent. Eyes are red and rather large. There is a steep punctuation on the mesothorax. Legs are brown. Male body is 4–6 mm long, antennae are long and hairy. Abdomen is longer than in the female and rounded at the back. Females are more persistent and about 7–7.5 mm long. Ovipositor is yellow. The egg is white, ovoid, with a long pedicel. The larva of the last instar is 6–7 mm long and legless. Body colour is white, cephalic capsule slightly yellow, and mandibles are brown. Larvae body is curved and sharp at both ends. Pupa is 4–7 mm long and the first stage is hyaline. During its transformation into the adult stage, the insect passes through the following developmental phases: a white pupa, a white pupa with pink eyes, a white pupa with pink eyes and developing legs, a partially darkened pupa with emerging legs and wing buds, and finally a pupa with fully developed black legs, wings, and antennae [1,6,8,9].

### 1.3. Life Cycle

In the climate of Romania, the southern part of Russia, Ukraine, and the Czech Republic, *E. schreineri* has one generation per year. The larvae mature overwinter in seeds of infested host plants. At the end of April, when the ambient temperature exceeds 10 °C, the larvae turn into pupae and when the temperature exceeds 15.6 °C, the pupae transform into adult insects [10]. Adults bore round holes 1.0–1.8 mm in diameter in the stone walls using their mandibles and fly away (Figure 1). After copulation, the female perforates the growing fruit pulp and lays eggs in the endosperm before the stone is formed (Figure 2). The female lays from 30 to 40 eggs in different plums. The life expectancy of adults is 6–8 days, maximum 15–18 days [10]. There may be two larvae in one fruit at the beginning, but only one larva completes development in the fruit. The larvae hatch from the egg after 16–22 days and consume the seed kernel (Figure 3). The larvae complete development during the summer and then enter diapause and hibernate in the fallen fruit, protected by strong stone walls, until the following spring [1,10,11].

### 1.4. Symptoms of Damage

Larvae feed inside of plum seeds. Attacked fruits are not visibly different from healthy ones, but they dry out and fall down in July (Figure 4). The skin becomes dry and adheres firmly to the stone. It looks like mummification. Locally, damage can be up to 90% [10]. The bare stones on the ground have a circular hole with a diameter of 1.0–1.8 mm (Figure 5) [12].

### 1.5. Control

Insecticide treatment must be applied to adults before females begin to lay eggs. The economic threshold level of the pest has been calculated as 1.2 larvae per tree in spring. The treatment has to be repeated two or three times, after 6–8 days [13], because the flying activity is often very protracted.

Currently, insecticides from the pyrethroid group (cypermethrin, alfa-cypermethrin, beta-cypermethrin, zeta-cypermethrin, deltamethrin, lambda-cyhalothrin, bifenthrin), organophosphates (chlorpyrifos, diazinon, fenitrothion, dimethoate), neonicotinoids (thiamethoxam, acetamiprid, thiacloprid), and spinosyns (spinosad) are registered in some European and Asian countries [6]. Most of these insecticides are not allowed for use on plums in many European countries, because of local public restrictions due to the eco-toxicological hazard level of these chemicals. The study by Moldovan [14] compared an organic treatment scheme using several natural products—such as paraffinic and vegetable oils (Ovipron Top, Oleorgan), *Bacillus thuringiensis* (Bactospeine DF), plant and seaweed extracts (Konflic, Deffort, Algasil), and orange oil (Prev-Am)—with conventional insecticides including acetamiprid, thiacloprid, lambda-cyhalothrin, thiamethoxam, and spirotetramat. While the conventional treatments reduced *E. schreineri* infestation on plum fruits to below 5%, the organic scheme resulted in a 12–28% damage on the susceptible ‘Stanley’ cultivar. The authors concluded that organic control alone is insufficient and should be complemented with tolerant cultivars, more frequent applications, and removal of infested fruits [14].

Removal of fallen fruits and stones is a non-insecticide recommended method for effective protection, as well as soil cultivation under the trees. The experiment made by Tertyshny [13] showed that adult *E. schreineri* cannot emerge from the soil when the stones are cultivated deeper than 5 cm. If the stones are buried in the soil to a depth of 10–15 cm using appropriate mechanization before the adult’s hatch, chemical treatment should not be necessary [13]. However, these methods are impractical or not applicable to large orchards. Deep tillage is not usually carried out under trees, and stone collection is not always possible in extensive orchards.

Another possibility of protection is the planting of resistant varieties. Resistance is conditioned by the firmness of the fruit stone when the adults fly and the thickness of the pericarp at the egg-laying time. If the stone is very hard, the adult cannot bite out the hole out and dies inside. The average length of the ovipositor is 2.7–2.9 mm. If the pericarp is too strong or thick, the female will not be able to lay eggs inside. *Eurytoma schreineri* is parasitised by *Syntomaspis eurytomae* (Puzanowa-Malysheva) from the Hymenoptera order and the Torymidae family. Its larvae first destroy the *Eurytoma* larva and then feed on the rest of the plum kernel. The imagoes of this parasite emerge 20–30 days later than the host and feed on nectar. About 1.6–3.4% of plum *Eurytoma* larvae were found to be infected with *S. eurytomae* in Ukraine [13].

## 2. Materials and Methods

### 2.1. Attractiveness of Colour Sticky Traps

The research on the attractiveness of visual traps for adults was carried out in the plum plantation (Czech Republic, 50.416619, 15.479180) with the predominant variety ‘Stanley’ in the years 2023 and 2024. Green, red, and blue traps were made of plastic plates (20 cm × 25 cm) sprayed with colour paint spray and painted on both sides with non-drying insect glue. Yellow and white traps, commercially available from Russel IPM^®^ (Deeside, UK), were used. In addition, two other types of yellow traps were added in 2024, the Rebell amarillo cross trap (Andermatt Biocontrol, Grossdietwil, Switzerland) and a yellow oval trap made from a 900 mL plastic bottle originally used for mustard (Novoplast, Moscow, Russia), painted with non-drying glue. Nine pieces of each type of trap were placed in 9 rows. One trap of a different colour or type was installed in each row. There was always 1 insulating row without traps between the rows with traps, and the insulation formed by 3 trees between the trees with traps in the row. Installation took place before the beginning of the plum blossom (30 April 2023 and 6 April 2024). The traps were hung in the treetops, about 1.5 m above the ground. Inspections and evaluations were conducted at approximately weekly intervals. Captured adults were counted and removed from the traps.

### 2.2. Residual Efficacy of Insecticides

Experiments were carried out under laboratory conditions with adult *E. schreineri*. Newly hatched adults were regularly collected from plastic boxes with plum stones removed from the infested orchard (Kamenice, Jičín district, GPS N 50°25.00572′, E 15°28.67913′).

In these experiments the residual efficacy of four agrochemicals in 2023 and five in 2024 was investigated. Distilled water was used as a control. The same number of wasps were counted per treatment in each experiment. 33 and 30 adult individuals were used in 2023 and 2024, respectively. The agrochemicals used in the study were selected based on their current (recently approved) and potential (likely to be approved in the future) use in plum orchards. All agrochemicals (Table 1) were tested at the concentrations and field doses recommended for orchards, according to the product labels, and while following the instructions provided of the Central Institute for Supervising and Testing in Agriculture (2023, 2024), which provides precise rules for the use of plant protection products.

For the residual efficacy test, each insecticide was sprayed into the Petri dish bottom (5.5 cm in diameter) and also into the inner part of the lid using a Potter precision laboratory spray tower (Burkard Scientific^®^, Uxbridge, UK). The standard application was between 1.5 and 2 mg cm^−2^, which corresponds to the wet deposition of pesticide for a volume of 400 L ha^−1^ in orchards. Each Petri dish and its lid were then allowed to dry under a laboratory hood. One *E. schreineri* individual adult was placed in each treated Petri dish along with a piece of moistened tissue to provide water. Each Petri dish was marked with a unique code consisting of an abbreviation for the treatment. Petri dishes were placed under laboratory conditions (22 ± 1 °C, 75 ± 5% relative humidity, 16:8 light/dark photoperiod). Mortality of each individual was recorded at 24, 48, 72, 96, and 120 h after treatment.

### 2.3. Contact Efficacy of Insecticides

For the contact efficacy test, each individual was sprayed using the Potter tower. The agrochemicals used and the amount of material applied were the same as for the residual efficacy method. The number of adult individuals used was 28 and 30 wasps per treatment in 2023 and 2024, respectively. Immediately after spraying, the adults were placed in clean Petri dishes (5.5 cm in diameter), and water was provided by a piece of moistened tissue as in the residual efficacy test.

### 2.4. Field Trial

The experiment started in 2024 in the plum orchard of a fruit growing company in Eastern Bohemia. A part of the orchard with a high density of *E. schreineri* population evenly distributed in the orchard was used for the experiment. In order to compare the effectiveness of the applied protection system, a part of the orchard was left untreated. The area of this part was about 1200 m^2^ (5 rows × 10 trees). The treatments were carried out on two dates with an interval of 7 days, mainly with regard to flight activity, pest development, and climatic conditions. The first application was made on 3 June 2024 with spinosad in a dose of 0.4 L ha^−1^. After 7 days, on 10 June 2024, acetamiprid was applied at the same orchard at a dose of 0.25 kg ha^−1^. The application was carried out with a pesticide tractor sprayer in 400 L ha^−1^ water volume. The experimental orchard with an area of approximately 0.51 ha was treated with the selected products acetamiprid and spinosad, the efficacy of which was proven in laboratory tests. Two treatments simulated the real need for application in relation to the prolonged flight activity of adult *E. schreineri.* We evaluated 5 × 100 (n = 500) plums from the treated and untreated parts of the orchard almost one month later, on 2 July 2024. These plums were opened with a vice and checked for the presence/absence of *E. schreineri* larvae.

### 2.5. Statistical Analysis

Prior to the comparative analysis, the data on the attractiveness of differently coloured sticky traps were tested for normality of residuals and homogeneity of variance using the exact Shapiro–Wilk and Cochran–Hartley–Bartlett tests, respectively. Due to a non-parametric distribution, the data were processed using the Wilcoxon–Mann–Whitney test. The results are described as median (Med.) with inter quartile range (IQR). The contact or residual efficacy of the insecticides in specific treatments was estimated from the survival curves of adult *E. schreineri* in each 24 h interval during the first five days. For these analyses, the survival curves were compared using the Kaplan–Meier (KM) model [15]. The KM model was used to estimate and visualize the differences between treatments at any given time during the observation period and to estimate the median survival time of *E. schreineri*. For the survival analysis, the data were uncensored, as all adults were evaluated throughout the time period without excluding any individuals. Differences between the treatments in the KM model are presented using *p*-values and confidence intervals (CI). The efficacy of the field treatment with acetamiprid and spinosad against *E. schreineri* was analyzed using the Chi-squared test comparing the proportion of stones engaged. The *p*-value < 0.05 was considered significant. All analyses were performed in the RStudio statistical software [16] working under R version 4.4.3. The following packages were used for these analyses: “survival” [17], “survminer” [18], “ranger” [19], “ggplot2” [20], “dplyr” [21], “ggfortify” [22] and “rlang” [23] packages.

## 3. Results

### 3.1. Attractiveness of Colour Sticky Traps

The yellow, green, and white sticky traps caught a higher number of *E. schreineri* (Med. 2.0–8.0 adults) than the red and blue traps (Med. 0 adults) in 2023. In that year, the IQR (interquartile range) was higher in the yellow sticky traps (Med. 24.5) compared to the other coloured traps (Med. 0.5–5.5) (Table 2).

In the following year 2024, the total number of adults captured was higher than in 2023. Particularly higher occurrence was observed on the yellow sticky traps (Med. 64.0 adults), the Rebell cross trap (Med. 46.0 adults), and the yellow bottle trap (Med. 18.0 adults) than in the other coloured traps (Med. 0.0–3.0 adults). The number of adults captured in the blue, white, and red coloured traps remained low, whereas the occurrence on the yellow traps increased significantly between 2023 and 2024. The IQR increased with the higher number of *E. schreineri* captured in 2024, showing high variability in the number of adults captured for each colour of the sticky traps (Figure 6), particularly on the yellow sticky desk, yellow Rebell cross, and yellow bottle trap.

### 3.2. Residual Effect of the Tested Agrochemicals

It is clear from the survival curves in Figure 7 that the median survival of the *E. schreineri* adults was the shortest when treated with spinosad in 2023. All treated adults were killed by the residue within the first 24 h showing 100% efficacy. With the use of thiacloprid, the median survival of adults was two days when applied as residue. However, after an initial efficacy of up to 40% within day one, the survival curve became more stable with decreasing mortality per day over the following days reaching approximately 80% efficacy compared to the control by the fifth day. A trend towards a residue effect was also observed for the 3D-IPNS polymers + silicones treatment. However, the difference from the control treatment was not significant except on the last day of the study. The efficacy of this product was only 35% compared to the control at the end of the trial and can be considered as low. Adult *E. schreineri* treated with spirotetramat had a survival curve similar to that of the control treatment.

The residue efficacy of the treatments in 2024 is shown in Figure 8. The results confirm the high efficacy of the spinosad treatment observed in 2023. This treatment showed 90% efficacy by killing the adults within the first 24 h. Similar results were obtained with sulfoxaflor and spinetoram active ingredients, where all these treatments achieved the median survival within the first 24 h and over 90% efficacy against the control. The preparation based on acetamiprid also reached high final efficacy, but treated adults had a significantly higher survival rate within the first two days when compared to those treated with spinosad, sulfoxaflor, and spinetoram. This suggests a slower onset of full efficacy of acetamiprid residue. Cyantraniliprole showed moderate to low residual activity with only a 31% reduction in adult survival compared to the control at the end of the trial. Although the survival time was significantly shorter than the control, this treatment did not reach the median survival line within the first five days. In comparison with the year 2023, the *E. schreineri* adults in the control treatment had already started to die after the second day of the trial, with up to 25% mortality at the end of the trial.

### 3.3. Contact Effect of the Tested Agrochemicals

In 2023, the contact efficacy of the products tested was similar to their residual efficacy. This was especially true for spinosad with 100% efficacy within the first day and spirotetramat with almost no efficacy (7%) by the end of the experiment compared to the control (Figure 9). However, for contact efficacy, thiacloprid showed only a 67% reduction in final survival rate compared to the control. The median survival of *E. schreineri* adults in this treatment was reached after three days, but the survival curve remained significantly different from that of the control treatment. The survival curve of the adults treated with 3D-IPNS polymers + silicones remained similar to the control.

The contact efficacy of the products tested in 2024 again showed significant differences (Figure 10). The best efficacy was observed after treatment with spinosad and spinetoram, where all adults were killed within the first 24 h. A similarly low median survival and therefore high efficacy (96%) within the first 24 h was also observed after treatment with acetamiprid. Adults treated with sulfoxaflor showed a significantly lower survival curve than the control, but the median survival was two days and full efficacy was achieved by the third day after the application. Cyantraniliprole does not appear to have any contact efficacy, as the survival curve of the adults treated with this product was similar to that of the control. The control treatment in 2024 was again characterized by a significantly higher mortality rate of up to 33% at the end of the trial.

### 3.4. Efficacy of Applied Protection System

The applied system of two insecticides on two dates within seven days was effective in reducing the proportion of infested fruit (Chi-squared = 322.69, df = 1, *p*-value < 2.2 × 10^−16^). As can be seen in Table 3, the proportion of plum stones infested by the *E. schreineri* in the treated part of the orchard was 0.22, while it was 0.79 of plum stones in the untreated part of the orchard.

## 4. Discussion

There are very few studies dealing with *E. schreineri*. Therefore, the discussions on this topic cannot be extensive.

### 4.1. Attractiveness of Colour Sticky Traps

There are no scientific studies available that examine the flight ability or the distance *E. schreineri* can cover. However, based on our experience in the orchard, this species is neither a strong nor a fast flyer, and that females in particular do not leave the hatching site. Males search for females using a pheromone, so their range could be greater. We are not aware of any experiments that have tested the attractiveness of visual traps for *E. schreineri*, which makes our results important. The results from both years of testing showed a significant preference for the yellow-coloured traps. Of the different types of yellow traps, most adults (866) were found on the oval type of trap, but 740 of the whole number were found just in two oval traps from nine pcs. Therefore, based on the results of the used statistics, the most effective trap was found to be the common yellow sticky trap.

### 4.2. Residual, Contact, and Field Efficacy Tests of Agrochemicals

The agrochemicals tested in this study were selected based on their current or recent registration and potential to likely be approved for orchards in the future. Spinosad, thiacloprid, silicone 3D-IPNS, and spirotetramat were tested in 2023. Silicone 3D-IPNS and spirotetramat were significantly ineffective and were discarded for subsequent experiments. Despite its relatively good efficacy, thiacloprid was also discontinued. Based on the Regulation (EU) 2020/23, the use of its active substance thiacloprid was phased out in the European Union in 2021. Excellent efficacy was achieved with spinosad in both types of experiments, i.e., contact and residual tests. Spinosad, spinetoram, sulfoxaflor, cyantraniliprole, and acetamiprid have been added to the tests as potentially effective insecticides in 2024. Spinetoram, sulfoxaflor, and spinosad showed the highest mortality from the first day after treatment in the residual evaluation. Treatment with acetamiprid resulted in high adult mortality, but the full efficacy of the product started after the third day. Treatment with cyantraniliprole resulted in a lower proportion of mortalities at the residual effect assessment. Spinosad, acetamiprid, and spinetoram were the most effective when the contact effect was assessed. Sulfoxaflor reached its highest efficacy after three days. The mortality caused by cyantraniliprole in the contact test was comparable to that recorded in the control treatment. For the majority of the tested products, no published scientific data were available for comparative evaluation in the discussion. Most of these products, although considered suitable for use in IPM programmes against the plum seed wasp, have not been previously evaluated in other studies.

The study made by Moldovan [14] evaluated two pest management systems—a conventional chemical programme and an organic scheme—for controlling *Eurytoma schreineri* in plum orchards using the cultivars ‘Stanley’ and ‘Reine Claude d’Althan’. The conventional programme included nine treatments with insecticides such as acetamiprid (Mospilan), thiacloprid (Calypso), lambda-cyhalothrin (Karate Zeon), thiamethoxam (Actara), and spirotetramat (Movento), applied in combination with fungicides (e.g., Dithane, Score, Topsin). This system effectively reduced infestation, with only 3–5% of fruits attacked on the sensitive cultivar ‘Stanley’ and about 1% on the more tolerant ‘Reine Claude d’Althan’ [14].

The organic scheme consisted of products with natural or biological origin, including Ovipron Top (paraffinic oil), Konflic (Quassia extract and potassium soap), Deffort (plant extract + microelements), Oleorgan (vegetable oils), Bactospeine DF (*Bacillus thuringiensis*), Algasil (seaweed extract with silicon and potassium), and Prev-Am (orange oil). Despite multiple applications, the organic treatments provided limited protection, with infestation levels of 12–28% on ‘Stanley’ and 3–4% on ‘Reine Claude d’Althan’ [14].

Overall, the study demonstrated that while conventional insecticides achieved effective control of *E. schreineri*, the organic scheme was insufficient to prevent economic losses, especially on susceptible cultivars. The authors recommend combining organic products with resistant varieties, frequent applications during the flight period, and sanitation measures such as removing and destroying infested fallen fruits [14].

Only in the experimental years 2008–2011, Rodica [24] tested the insecticide Calypso 480 SC (0.2%). Efficacy varied between 93.5% and 94.50%. Some studies have focused on testing of insecticides that are not allowed in IPM. Tălmaciu [25] tested the efficacy of Decis 2.5 EC (deltamethrin) 0.05% with the frequency of attacked fruit of 2.0% (efficacy 98%), Karate 2.5 EC (lambda-cyhalothrin) 0.03% with the frequency of attacked fruit of 0.8% (efficacy 99.2%), the biological product Dipel (*Bacillus thuringiensis* subsp. kurstaki) 0.1% with the frequency of attacked fruit of 21% (efficacy 89%), and Bactospeine (*Bacillus thuringiensis*) with the frequency of attacked fruit of 19% (efficacy 81%). Arnaudov [10] carried out preliminary tests with the contact insecticide Karate Zeon (lambda-cyhalotrine) at a dose of 200 mL ha^−1^ evaluating its efficacy against adults. He tested two strategies: The first was a single treatment applied only at the beginning of emergence, and the second strategy was a second treatment applied two weeks after the first one. Their data showed that the protection of plum orchard against *E. schreineri* with a single application of Karate is satisfactory. Rodica [24] also tested Actara 25 WG (thiamethoxam) 0.01%, with an efficacy of 94.50% and Regent 200 SC (fipronil) 0.01%, with an average efficacy of 92.03%. The lowest efficacy was reported in the group of synthetic pyrethroid insecticides, Cyperguard 25 EC (cypermethrin) 0.02% with an efficacy of 88.63% and Decis 25 WG (deltamethrin) 0.025% with an efficacy of 88.85%. The insecticide Talstar 10 EC (bifentrin) 0.04% caused a mortality of 90.10%. The main reason for testing the new products described above was the fact that these chemical insecticides from the pyrethroid group are not approved for use in IPM in fruit orchards in most European countries. The results of our experiments are therefore applicable in many countries.

## 5. Conclusions

Although very little was known about the *E. schreineri*, the present study has shed light on many new insights which can be useful for the innovation of integrated pest management. This study summarizes the knowledge about *E. schreineri,* updates the possibilities of monitoring adults, and the protections against this pest. Trials showed that yellow sticky traps were the most attractive among other coloured traps—red, green, blue, and white. Spinosad was the most effective active ingredient in all laboratory trials. Its efficacy was also confirmed under field conditions. These new insights into the efficacy of the products tested will help growers in their selection of active substances when implementing protective measures.

## Figures and Tables

**Figure 1 insects-16-01112-f001:**
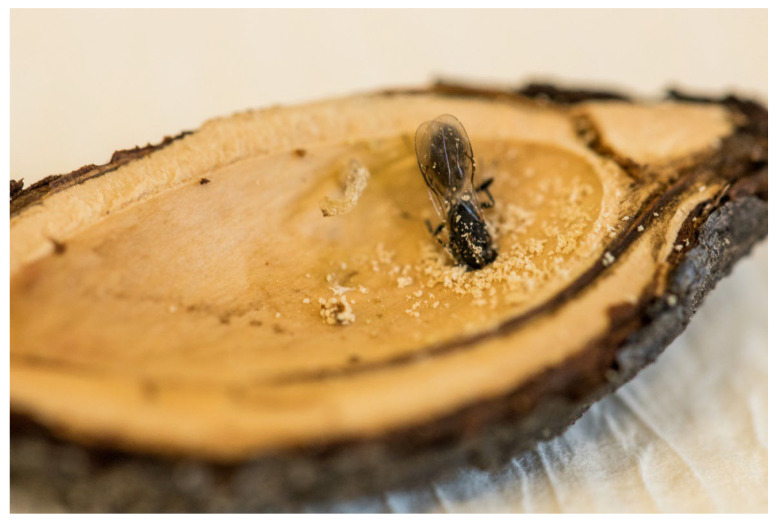
Adult of *E. schreineri* bite through the stone.

**Figure 2 insects-16-01112-f002:**
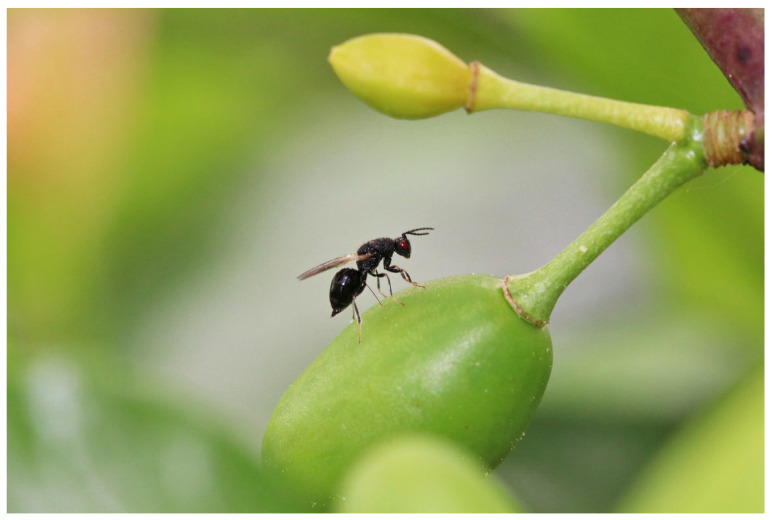
*E. schreineri* female preparing to lay eggs in a plum.

**Figure 3 insects-16-01112-f003:**
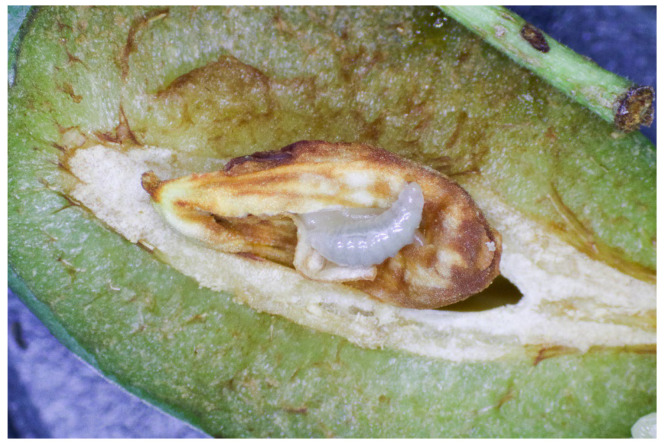
Young larva of *E. schreineri* feeds on kernel inside plum.

**Figure 4 insects-16-01112-f004:**
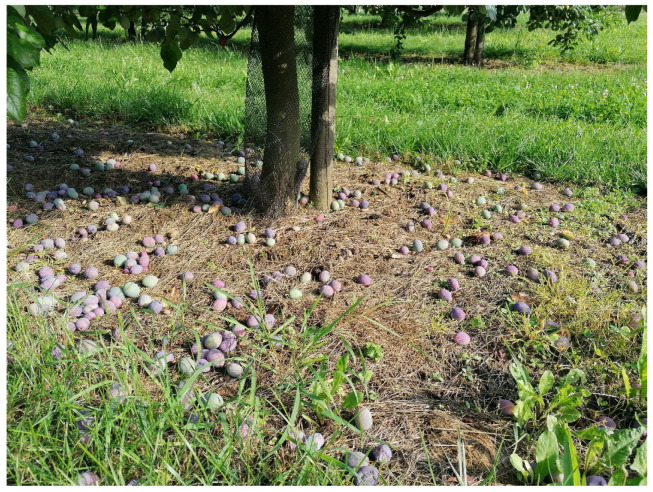
Dropped plums relating to *E. schreineri* damage; larvae are inside plums where they overwinter.

**Figure 5 insects-16-01112-f005:**
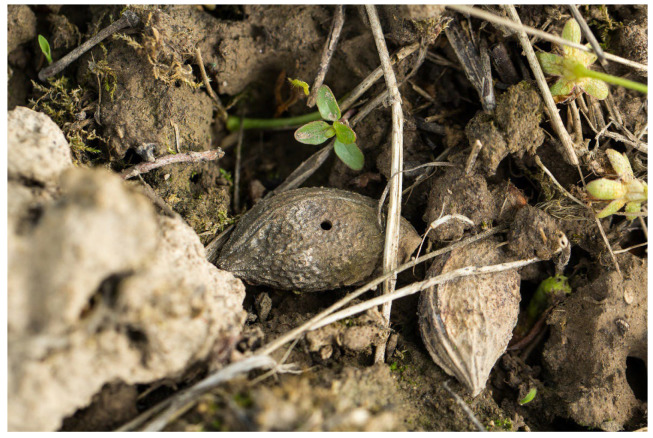
Old plum stone with hole from adult leaving.

**Figure 6 insects-16-01112-f006:**
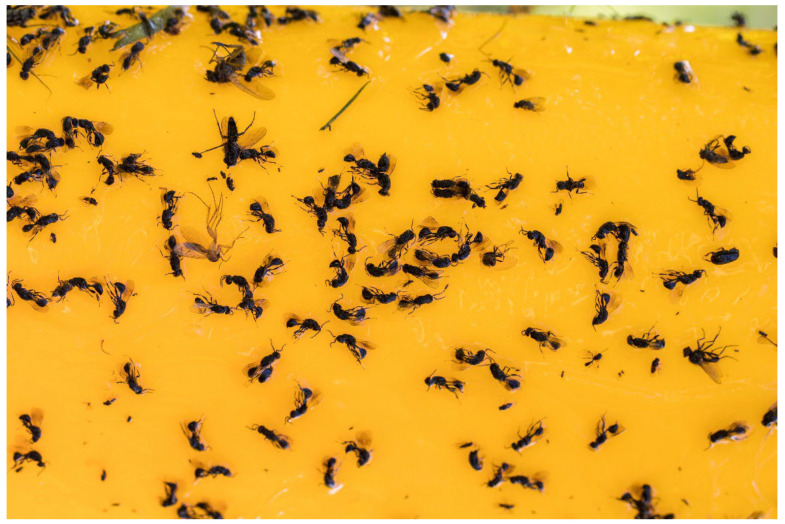
*E. schreineri* adults caught on yellow sticky trap as tool for monitoring.

**Figure 7 insects-16-01112-f007:**
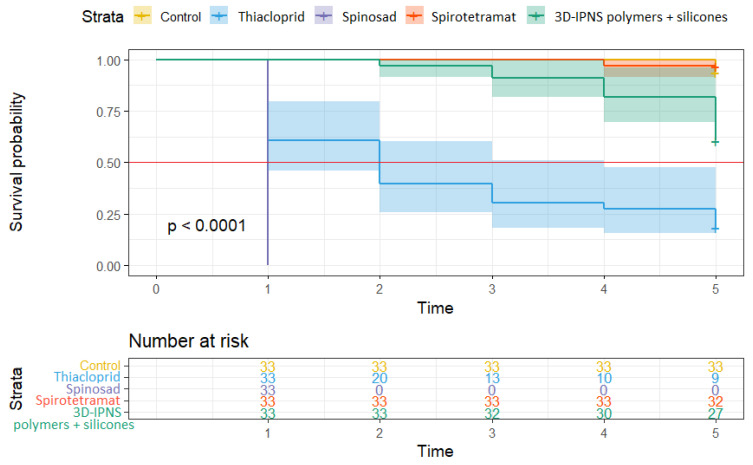
The Kaplan–Meier curves comparing survival estimates between the groups of *E. schreineri* adults treated with specific insecticides evaluated for residue efficacy in 2023. Differences are visualized using CI. The median survival borderline is shown by the red line.

**Figure 8 insects-16-01112-f008:**
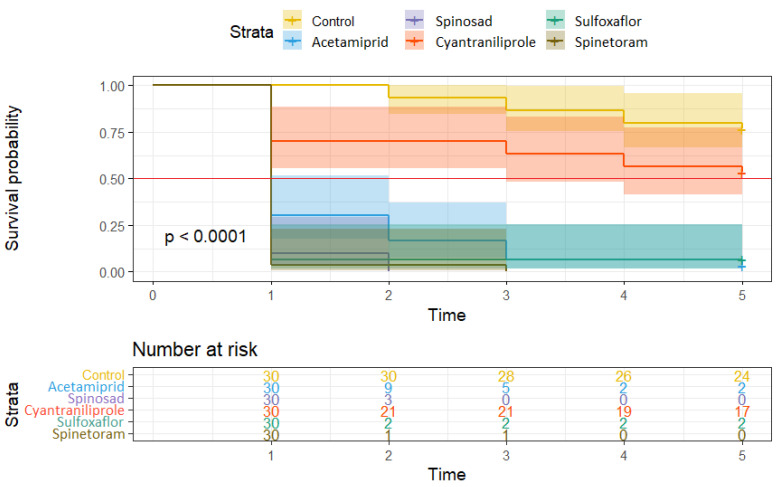
The Kaplan–Meier curves comparing survival estimates between the groups of *E. schreineri* adults treated with specific insecticides evaluated for residue efficacy in 2024. Differences are visualized by CI. The median survival borderline is shown by the red line.

**Figure 9 insects-16-01112-f009:**
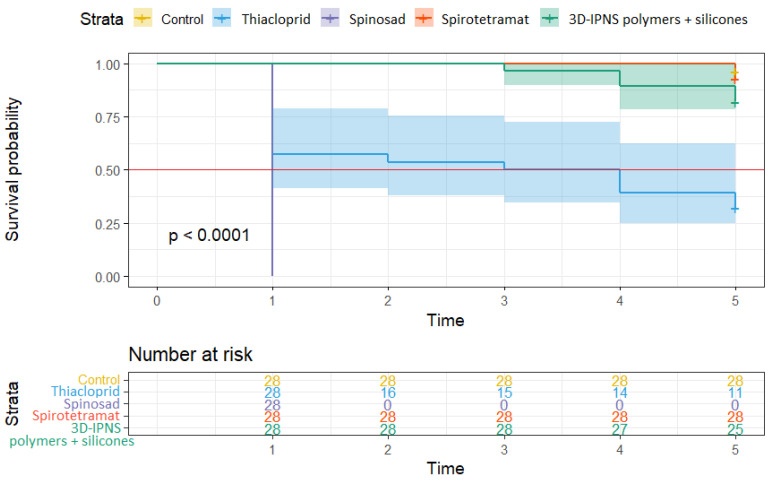
The Kaplan–Meier curves comparing survival estimates among the groups of *E. schreineri* adults treated with specific insecticides evaluated for contact efficacy in 2023. Differences are visualized using CI. The median survival borderline is shown by the red line.

**Figure 10 insects-16-01112-f010:**
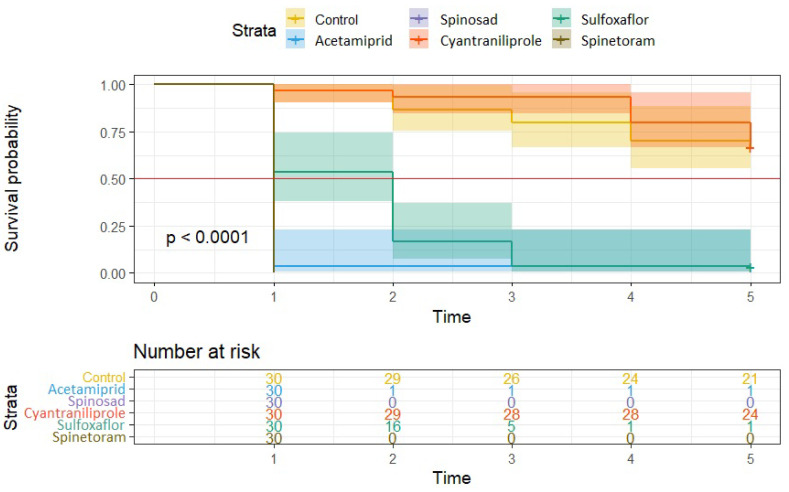
The Kaplan–Meier curves comparing survival estimates among the groups of *E. schreineri* wasps treated with specific insecticides evaluated for contact efficacy in 2024. Differences are visualized using CI. The median survival borderline is illustrated by the red line.

**Table 1 insects-16-01112-t001:** Characteristics of the pesticides used in the studies to investigate possible effects on *E. schreineri*.

Active Ingredient	Trade Name and Formulation	Dose ha^−1^	Year of Testing	Registrant
Thiacloprid (480 g L^−1^)	Calypso 480 SC^®^	0.25 L	2023	Bayer AG (Leverkusen, Germany)
Spinosad (240 g L^−1^)	SpinTor^®^	0.4 L	2023 2024	Dow AgroSciences Ltd. (Indianapolis, IN, USA)
Spirotetramat (100 g L^−1^)	Movento 100 SC^®^	1.5 L	2023	Bayer S.A.S. (Lyon, France)
3D-IPNS polymers + silicons	Siltac EC^®^	1.5 L	2023	ICB Pharma (Monheim, Germany)
Acetamiprid (200 g L^−1^)	Mospilan 20 SP^®^	0.25 kg	2024	Nisso Chemical Europe GmbH (Düsseldorf, Germany)
Cyantraniliprole (100 g L^−1^)	Benevia^®^	0.75 L	2024	FMC Agro (Prague, Czech Republic)
Spinetoram (120 g L^−1^)	Radiant SC^®^	0.40 L	2024	Dow AgroSciences Ltd. (Indianapolis, IN, USA)
Sulfoxaflor (120 g L^−1^)	Gondola^®^	0.20 L	2024	Dow AgroSciences Ltd. (Indianapolis, IN, USA)

**Table 2 insects-16-01112-t002:** Comparison of colour attractiveness of sticky traps for *E. schreineri* in plum orchard in years 2023–2024.

Year	Type of Trap	Median	IQR
2023	Blue sticky desk	0.0 b	0.5
	Green sticky desk	3.0 a	4.5
	Red sticky desk	0.0 b	2.0
	White sticky desk	2.0 a	5.5
	Yellow sticky desk	8.0 aB	24.5
	Yellow Rebell cross trap	-	-
	Yellow bottle trap	-	-
2024	Blue sticky desk	0.0 c	2.0
	Green sticky desk	3.0 b	20.0
	Red sticky desk	1.0 bc	10.5
	Yellow sticky desk	64.0 aA	87.5
	White sticky desk	0.0 bc	2.5
	Yellow Rebell cross trap	46.0 a	111.0
	Yellow bottle trap	18.0 a	127.0

Significant differences in the occurrence of captured *E. schreineri* between the sticky traps of different colours for particular years are presented as median values with different lowercase letters at *p* < 0.05. Differences in the occurrence of adults on the sticky traps of the same colour between the observation years are indicated by different capital letters at *p* < 0.05.

**Table 3 insects-16-01112-t003:** The resulting amount of *E. schreineri* infested/not infested plum stones.

Treatment	Replication	Infested Plums	Not Infested Plums
Treated orchard	A (n = 100)	28	72
	B (n = 100)	10	90
	C (n = 100)	7	93
	D (n = 100)	35	65
	E (n = 100)	31	69
	Total (n = 500)	111	389
Untreated orchard	A (n = 100)	70	30
	B (n = 100)	92	8
	C (n = 100)	83	17
	D (n = 100)	73	27
	E (n = 100)	78	22
	Total (n = 500)	396	104

## Data Availability

The original contributions presented in this study are included in the article. Further inquiries can be directed to the corresponding author.

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
