# Peer review of "A New Invasive Pest in Plums: Monitoring and Control of Eurytoma schreineri"

_insects, 2025, doi:10.3390/insects16111112_

Round 1
Reviewer 1 Report
Comments and Suggestions for Authors
The manuscript is informative and provides a wealth of data that is currently missing in the literature. Therefore the discussion section is short.
There are aspects that need to be explained more thoroughly to provide greater clarity and precision regarding the applied methods, particularly in the analysis of the results. Some sections lack clarity and appear illogical.
All comments are written in the document attached.

English needs improvement in certain parts of the text because the meaning is not clear enough.
It can be speculated about the authors' intentions, but that is insufficient for publication in a scientific journal.
Author Response
Dear reviewer,
thank you for the time you devoted to reviewing our paper. We appreciate your comments and suggestions for revisions, which we have tried to incorporate into the manuscript according to your recommendations.
Best regards, Michal Skalsky.
In Introduction section, lines 24-32:
Comment 1 - after mentioning the Latin name of the insect for the first time, I suggest to add order and family in the brackets, just to inform readers which are not familiar with wasps
Response 1 - Corrected
Comment 2 - next, when you start sentence with Latin name of the species (line 25 two times) you have to write the whole name, not short version, although you already mentioned it. The sentence can’t start with one letter and full stop.
Response 2 - Corrected
Comment 3 - in line 26, put Latin name of the plum in italic letters
Response 3 - Corrected
Comment 4 - in line 27 it is unusual to start citations from the 20th reference on the list. it is common the put reference numbers in order of appearance (“References must be numbered in order of appearance in the text (including table captions and figure legends) and listed individually at the end of the manuscript” – citation from the instructions to authors on Insects web page).
Response 4 - Corrected
Comment 5 - Line 29 delete the second ‘also’, you repeat the same word twice in the same sentence
Response 5 - Corrected
Comment 6 - line 31 delete ‘therefore’
Response 6 - Corrected
Comment 7 - line 34, you can’t start a whole paragraph with “It”. Be precise what do you mean by IT.. at the beginning of the sentence and paragraph also; use something like the wasp, the species.. or something like that
Response 7 - Corrected
Comment 8 - line 35 add preferred host species
Response 8 - Corrected
Comment 9 - line 36: after blackthorn add comma, cause you mention its Latin name, the same plant
Response 9 - Corrected
Comment 10 - line 37 put P. avium in italic. Also, in the whole paragraph 1.1 use short version of genus Prunus after first time mentioned species name.
Response 10 - Corrected
Comment 11 - line 38 after sour cherry add comma
Response 11 - Corrected
1.2. Insect description
Comment 12 - line 40, the same comment as previous, don’t start the whole paragraph with one letter and full stop, add the full genus name at the beginning of sentence
Response 12 - Corrected
Comment 13 - lines 48-51: “By transforming the adult goes through several stages: white 48 nymph, white nymph with pink eyes, white nympha with 230 pink eyes and feet initiated, 49 nympha half black with legs and wings initiated, nymph with black legs, wings and an-50 tennae initiated, nymph with black legs, wings and antennae released” – this sentence is absolutely not clear, it must be rephrase to get better clarity. I also suggest to divided it in at least two sentences, describing of preimaginal stages and pupa and adult, e.g. What does it mean 230 pink eyes?!?
Response 13 - Corrected
Comment 14 - “By transforming the adult goes…” what exactly do you mean by that?! I can imagine you want to say that the specimen is transformed through different life stages, but as you wrote it has no sense!
Response 14 - Corrected
Comment 15 - Line 56 where you mention temperatures, you must add citation, where did you get this information. It is not enough to put few citations at the end of paragraph. There must be an author or authors who defined the mentioned conditions, they deserve to be mentioned.
Response 15 - Added
Comment 16 - Line 57 you wrote that Adults bore round holes, describe how they do it, by cewing it with their mouth parts or using ovipositor, it’s not the same..
Response 16 - Corrected
Comment 17 - Line 60, again you mention life span of adults without any citation..
Response 17 - Corrected
Comment 18 - Line 63. How can larvae ‘..complete development during the summer diapause…’ If they diapauses they can’t finish their life cycle, there is no logic, please explain better the whole process, be more precise. Is summer diapauses related to insect or host plant?
Response 18 - Corrected
Comment 19 - Line 69 you wrote: “Figure 2. E. schreineri female prepare to laying egg on plum.” change laying to LAY egg
Response 19 - Corrected
Comment 20 - Liines 83-84: you put figure 6 before mentioning. Please change that.
Response 20 - Corrected
Comment 21 - Line 94 delete’there’ befor citation
Response 21 - Corrected
Comment 22 - Line 98 you wrote..” well as cultivation soil under the trees.”, cultivation soil change to soil cultivation
Response 22 - Corrected
Comment 23 - Line 99 citation “Tertyshny [22] showed” is not numbered appropriate
Response 23 - Corrected
Comment 24 - Line 109 again don’t start sentence with short Latin name.. as on the beginning of text
Response 24 - Corrected
Comment 25 - Line 110 Latin name Syntomaspis eurytomae pit in italic letters, also genus Eurytoma
Response 25 - Corrected
Comment 26 - Lines 112-113 again, put Latin names in italic
Response 26 - Corrected
Comment 27 - In paragraph 1.5. Control you cited only one paper, numbered 22 (check numbering!) and only once 18. I suggest to add few more citations, in order to show best controlling practices. In your MS citation number 22 is related to both chemical and biological treatments (even the mechanical measures..), try to find more related papers to cite
Response 27 - Corrected
Comment 28 - Line 116 you wrote: “…if there existed..”, correct English here, this is not acceptable; I guess you mean if there exists an optical trap..
Response 28 - Corrected
Comment 29 - Lines 114-120 this paragraph doesn’t belong to the previous subtitle Control. On the other hand, if one of the aims of your work was to find out the appropriate trap for monitoring, then your introduction part must have this explanation also. With references and known facts about it.
Response 29 - Corrected
Comment 30 - Line 116, line 124.. are you sure you are searching for optical trap? I would call it visual trap.
Response 30 - Corrected
Comment 31 - Line 125 in M&M section, where the research was conducted?
Response 31 - Corrected
Comment 32 - Line 152 you wrote: “..each treatment was sprayed..”, I guess you mean each insecticide/agrochemical, the treatment can’t be sprayed
Response 32 - Corrected
Comment 33 - Line 157: you wrote “One individual was placed..”, be precise which individual in which life stage (larva, adult..)
Response 33 - Corrected
Comment 34 - Lines 179-180 you wrote: “After 7 days, on 10 June 2024, acetamiprid was applied at 179 a dose of 0.25 kg ha-1.”, was the second treatment conducted on the same plot/part of orchard? You have to underline that.
Response 34 - Corrected
Comment 35 - Line 190 you wrote: “…sticky traps were tested for normality of residuals and homogeneity of variance..”, can you rephrase this for more clarity?
Response 35 - Corrected
Comment 36 - Lines 208-209: you wrote: The yellow, green and white sticky traps caught a higher number of E. schreineri captured than the red and blue traps in 2023. Please rephrase for better clarity. I would delete ‘captured’
Response 36 - Corrected
Comment 37 - In Table 2. you should underline if the letters for both years are the same or not. For example: Green sticky desk in 2023 with Median value of 3.0 is marked with a, while in 2024 with the same M value 3.0 is marked with b. With b is also marked Blue sticky desk in 2023 with value 0.0. How do you explain that?
Response 37 - Corrected
Comment 38 - lines 291-293 you wrote: “the proportion of plum stones infested by the E. schreineri in the treated part of the orchard was 0.22 while it was only 0.79 of plum stones in the untreated part of the orchard”, how did you get this proportion? Why 0.79 is ‘only’ 0.79 if the proportion of infesed and non infested plums was far better in treatment than in control? Give more clarity to the explanation of results.
Response 38 - Corrected
Comment 39 - Line 299-300 you wrote: “It is known that E. schreineri is not a good flyer and that females in particular do not 299 leave the hatching site.”. who knows, based on which reference?! You must add some reference for this ‘known’ fact
Response 39 - Added
Comment 40 - line 301 again, optical or visual trap?
Response 40 - Corrected
Comment 41 - line 306 you wrote: “The number of caught adults was influenced by the focal occurrence.”, based on what you clamed this?
Response 41 -
Comment 42 - Line 316-317 you wrote:” the use of its active substance thiacloprid will be phased out in the European Union in 2021”. You write the paper in 2025 and use future past tense for something that happened in 2021. This sentence needs revision. Please adjust tenses accordingly!
Response 42 - Corrected
Comment 43 - Line 326 you wrote: “Cyantraniliprole mortality..”, this must be rephrased, cyantraniliprole can’t have mortality, it can cause mortality of specimens. Please rephrase.
Response 43 - Corrected
Comment 44 - Line 331: You wrote: “[21] tested the efficacy 331 of Decis 2.5 EC (deltamethrin) 0.05% with the frequency of attacked fruit 2.0% ..” You must start sentence with name or some word, not with brackets! Line 336 the same! line 341,
Response 44 - Corrected
Comment 45 - Line 346 add full stop before The..
Response 45 - Corrected
Comment 46 - Line 352 Latin name in italic, line 355, 357,
Response 46 - Corrected
Comment 47 - Line 378 you have two numbers “3. Ayaz..”
Response 47 - Corrected
Reviewer 2 Report
Comments and Suggestions for Authors
Dear Editor:
The manuscript entitled: A new invasive pest in plums: “Monitoring and control of Eurytoma schreineri” is an interesting work in which the authors propose to analyze the trapping and chemical control as two important techniques for the management of E. schreineri a pest of plums. They evaluated some types of traps and insecticides for the monitoring and suppression of pest population. This is basic information to start a program of pest control, however, the manuscript need to improve significantly the presentation.
In my opinion this manuscript must be continue in review with major changes.
At follow I included some specific and general comments, which can help to improve the presentation of the manuscript.
L-26 Prunus domestica in italics.
L-109 Syntomaspis eurytomae in italics, and it is important to include the taxonomic descriptor.
The introduction has important information about the pest and its dispersion. However, the authors should do more delimitations of the objectives. They only did mention at the final part of the introduction. Besides, the objectives are very different, I recommend to analyze if it would be better to include a general objective in which the monitoring and chemical control of the pests are the main goal.
L-202 It is more clear Chi-square test.
The section of results should be good to include the statistic parameter obtained from each test.
I considered that the data obtained from trapping and insecticide evaluations should be analyzed included other variables, for example there were differences among year of sampling, and maybe environmental conditions.
Other important comment, there are so figures, if the data became better organized in a complete design, could be more easy to do the presentation.
The discussion must be supported with references.
This manuscript describes important information, however, I think that it is very important to adequate the follow:
The introduction must be more closed with the definition of the objectives.
The objectives also, must be more integrative. There are two technical important subjects, the trapping and the insecticides.
The design of the evaluations must include other variables as the time, and possibly environmental conditions.
This could be important to have more options to introduce the results, avoiding so many figures.
Also, the discussion could be more complete and the conclusion could also have stronger information.
Author Response
Dear reviewer,
thank you for the time you devoted to reviewing our paper. We appreciate your comments and suggestions for revisions, which we have tried to incorporate into the manuscript according to your recommendations.
Best regards, Michal Skalsky.
- L-26 Prunus domestica in italics - corrected
- L-109 Syntomaspis eurytomae in italics, and it is important to include the taxonomic descriptor - corrected
- The introduction has important information about the pest and its dispersion. However, the authors should do more delimitations of the objectives. They only did mention at the final part of the introduction. Besides, the objectives are very different, I recommend to analyze if it would be better to include a general objective in which the monitoring and chemical control of the pests are the main goal - The objectives were stated more clearly in the introduction.
- L-202 It is more clear Chi-square test - updated
- The section of results should be good to include the statistic parameter obtained from each test. In the Methodology chapter, section 2.5 Statistical analysis, additional data have been added. Data in the Results chapter have also been specified in more detail, particularly in section 3.4. If necessary, please provide a more detailed clarification of your comment.
- I considered that the data obtained from trapping and insecticide evaluations should be analyzed included other variables, for example there were differences among year of sampling, and maybe environmental conditions. The difference between the years 2023 and 2024 in testing the attractiveness of visual traps was described. The laboratory experiments were conducted under the same conditions each time. The field experiment was carried out in only one year.
- Other important comment, there are so figures, if the data became better organized in a complete design, could be more easy to do the presentation. A separate graph was created for each experiment, because each experiment had its own control variant. For better clarity, each year is presented separately.
- The discussion must be supported with references - updated
This manuscript describes important information, however, I think that it is very important to adequate the follow:
- The introduction must be more closed with the definition of the objectives – added
- The objectives also, must be more integrative. There are two technical important subjects, the trapping and the insecticides – added (Introduction) The main objective was to build up methodology for monitoring and control against E. schreinery. This was achieved by determining of most attractive visual trap as a simple tool to determine the presence, abundance, and optimal duration of treatment against E. schreineri adults. Another goal was to find an effective insecticidal treatment allowed in IPM that would prevent damage caused by E. schreineri. Insecticidal treatment against adults of E. schreineri is currently the main option for protecting plums from this dangerous invasive pest.
- The design of the evaluations must include other variables as the time, and possibly environmental conditions. Monitoring of environmental conditions was not the main objective of this study. The objectives are now stated in the introduction. More in responses 6 and 10.
- This could be important to have more options to introduce the results, avoiding so many figures. More details are provided in response 7.
- Also, the discussion could be more complete and the conclusion could also have stronger information - added
Round 2
Reviewer 1 Report
Comments and Suggestions for Authors
All comments referred to version 2 of the manuscript are given as below.
Again, the description of life stages is unacceptable.
The rest can be easily changed/improved.
Lines 57-61 you wrote: “Male body is 4-6 mm long, antennae are long and hairy. Abdomen is longer than in female, rounded at the back. Females are more persistent and about 7-7.5 mm long. Ovipositor is yellow. The egg is white, ovoid, with a long pedicel. The larva of last instar is 6-7 mm long and legless. Body colour is white, cephalic capsule slightly yellow and mandibles are brown. Larvae body is curved and sharp at both ends. Pupa is 4-7 mm long and the first stage is hyaline. During its transformation into the adult stage, the insect passes through the following developmental phases: a white nymph, a white nymph with pink eyes, a white nymph with pink eyes and developing legs, a partially darkened nymph with emerging legs and wing buds, and finally a nymph with fully developed black legs, wings, and antennae. [1, 6, 8, 9, 10].”.
Again it is not clear and confusing, please put text in order. Hymenoptera have 4 life stages: egg, larva, pupa and adult. You start explanation with adults, like male body…females.. then you mention egg is white… then larva….pupa (which goes to adults again). And after explanation of stages (bold in above text) you wrote ‘During its transformation into the adult stage, the insect passes through the following developmental phases..’. during transformation of what? which stage? (‘its transformation’ is related to which stage exactly?) And how after explanation of larvae you came to nymphs?! Nymph is usually expression used for preimaginal stages of heterometabolous insects. Hymenoptera are holometabolous. If certain larval instars have different names, then you should say they are called hyaline.. or something like that. You already explained life stages, and then you finish text with explanation of ‘..finally a nymph with fully developed …wings!..’ there is no nymph with developed wings in insects!
The life stages are absolutely not clear and this text (lines 52-61) needs revision again. Delete dot before citations in brackets.
line 68 use the same font as the rest of text
Line 74 delete to in “..enter TO diapauses..”
Line 105 citation Moldovan (2020) is not cited correctly, the number off citation is missing
Line 110 use short version of the species name, you already mentioned it
Line 116 added words need space between
Line 131 use short name version for Syntomaspis eurytomae
Line 325 use short version of the species name, you already mentioned it
Line 332 add WERE between ‘number and found’
Lines 335-336 you wrote: “When statistical methods were used, the most effective trap (with more regular capture per trap) was found to be the common yellow sticky trap”. I would not say when statistical methods were used…results were this and that… These results don’t depend on statistical methods, you should say based on used statistical methods the most effective trap was the yellow sticky trap. I don’t understand this part in bracket that has been added, what’s the purpose of adding it..
Line 358 Moldovan 2020 is not properly cited
Line 379 Rodica (2012) [24] is not properly cited, Line 381 again.. Talmaciu, delete year, line 392
Line 403 add full stop at the end of sentence
Reference number 8 is cited where you explain life stage of the wasp. How exactly British Journal of Cancer is related to this topic?
Reference number 10. Are you sure you read reference from year 1908? I am not sure if this is cited properly, please check instructions for authors.
reference number 17 I think it is not properly cited. Is it a journal of web page?!
Author Response
Dear, thank you for your further comments and recommendations on our article, which we have incorporated according to your suggestions. Best regards, Michal Skalsky.
Comment 1 - Lines 57-61 you wrote: “Male body is 4-6 mm long, antennae are long and hairy. Abdomen is longer than in female, rounded at the back. Females are more persistent and about 7-7.5 mm long. Ovipositor is yellow. The egg is white, ovoid, with a long pedicel. The larva of last instar is 6-7 mm long and legless. Body colour is white, cephalic capsule slightly yellow and mandibles are brown. Larvae body is curved and sharp at both ends. Pupa is 4-7 mm long and the first stage is hyaline. During its transformation into the adult stage, the insect passes through the following developmental phases: a white nymph, a white nymph with pink eyes, a white nymph with pink eyes and developing legs, a partially darkened nymph with emerging legs and wing buds, and finally a nymph with fully developed black legs, wings, and antennae. [1, 6, 8, 9, 10].”.
Again it is not clear and confusing, please put text in order. Hymenoptera have 4 life stages: egg, larva, pupa and adult. You start explanation with adults, like male body…females.. then you mention egg is white… then larva….pupa (which goes to adults again). And after explanation of stages (bold in above text) you wrote ‘During its transformation into the adult stage, the insect passes through the following developmental phases..’. during transformation of what? which stage? (‘its transformation’ is related to which stage exactly?) And how after explanation of larvae you came to nymphs?! Nymph is usually expression used for preimaginal stages of heterometabolous insects. Hymenoptera are holometabolous. If certain larval instars have different names, then you should say they are called hyaline.. or something like that. You already explained life stages, and then you finish text with explanation of ‘..finally a nymph with fully developed …wings!..’ there is no nymph with developed wings in insects!
The life stages are absolutely not clear and this text (lines 52-61) needs revision again. Delete dot before citations in brackets.
Response 1 - It is true that the term nymph was used incorrectly and could be confusing. It actually refers to the pupal phases that the pest goes through inside the stone before becoming an adult. We have revised the text to make this clear.
Comment 2 - line 68 use the same font as the rest of text
Response 2 - Corrected
Comment 3 - Line 74 delete to in “..enter TO diapauses..”
Response 3 - Corrected
Comment 4- Line 105 citation Moldovan (2020) is not cited correctly, the number off citation is missing
Response 4 - Corrected
Comment 5 - Line 110 use short version of the species name, you already mentioned it
Response 5 - Corrected
Comment 6 - Line 116 added words need space between
Response 6 - Corrected
Comment 7 - Line 131 use short name version for Syntomaspis eurytomae
Response 7 - Corrected
Comment 8 - Line 325 use short version of the species name, you already mentioned it
Response 8 - Corrected
Comment 9- Line 332 add WERE between ‘number and found’
Response 9 - Corrected
Comment 10 - Lines 335-336 you wrote: “When statistical methods were used, the most effective trap (with more regular capture per trap) was found to be the common yellow sticky trap”. I would not say when statistical methods were used…results were this and that… These results don’t depend on statistical methods, you should say based on used statistical methods the most effective trap was the yellow sticky trap. I don’t understand this part in bracket that has been added, what’s the purpose of adding it..
Response 10 - Corrected
Comment 11 - Line 358 Moldovan 2020 is not properly cited
Response 11 - Corrected
Comment 12 - Line 379 Rodica (2012) [24] is not properly cited, Line 381 again.. Talmaciu, delete year, line 392
Response 12 - Corrected
Comment 13 - Line 403 add full stop at the end of sentence
Response 13 - Corrected
Comment 14 - Reference number 8 is cited where you explain life stage of the wasp. How exactly British Journal of Cancer is related to this topic?
Response 14 - Corrected
Comment 15 - Reference number 10. Are you sure you read reference from year 1908? I am not sure if this is cited properly, please check instructions for authors.
reference number 17 I think it is not properly cited. Is it a jornal of web page?!
Response 15 - Corrected
Reviewer 2 Report
Comments and Suggestions for Authors
In my opinion the current version has been improved substantially. My suggestions is to accept this versio to publish in Insects.
Author Response
Dear reviewer.
thank you for your time.
All the best, Michal Skalsky.